# Direct Patlak Reconstruction of [^68^Ga]Ga-PSMA PET for the Evaluation of Primary Prostate Cancer Prior Total Prostatectomy: Results of a Pilot Study

**DOI:** 10.3390/ijms241813677

**Published:** 2023-09-05

**Authors:** Sazan Rasul, Barbara Katharina Geist, Holger Einspieler, Harun Fajkovic, Shahrokh F. Shariat, Stefan Schmitl, Markus Mitterhauser, Rainer Bartosch, Werner Langsteger, Pascal Andreas Thomas Baltzer, Thomas Beyer, Daria Ferrara, Alexander R. Haug, Marcus Hacker, Ivo Rausch

**Affiliations:** 1Department of Biomedical Imaging and Image-Guided Therapy, Division of Nuclear Medicine, Medical University of Vienna, 1090 Vienna, Austria; sazan.rasul@meduniwien.ac.at (S.R.); barbara.geist@meduniwien.ac.at (B.K.G.); holger.einspieler@meduniwien.ac.at (H.E.); stefan.schmitl@akhwien.at (S.S.); markus.mitterhauser@meduniwien.ac.at (M.M.); rainer.bartosch@meduniwien.ac.at (R.B.); werner.langsteger@meduniwien.ac.at (W.L.); alexander.haug@meduniwien.ac.at (A.R.H.); marcus.hacker@meduniwien.ac.at (M.H.); 2Department of Urology, Comprehensive Cancer Center, Vienna General Hospital, Medical University of Vienna, 1090 Vienna, Austria; harun.fajkovic@meduniwien.ac.at (H.F.); shahrokh.shariat@meduniwien.ac.at (S.F.S.); 3Department of Urology, Weill Cornell Medical College, New York, NY 10065, USA; 4Department of Urology, Second Faculty of Medicine, Charles University, 15006 Prague, Czech Republic; 5Institute for Urology and Reproductive Health, I.M. Sechenov First Moscow State Medical University, 119991 Moscow, Russia; 6Department of Urology, University of Texas Southwestern Medical Center, Dallas, TX 75390, USA; 7Department of Biomedical Imaging and Image-Guided Therapy, Division of General and Pediatric Radiology, Medical University of Vienna, 1090 Vienna, Austria; pascal.baltzer@meduniwien.ac.at; 8QIMP Team, Center for Medical Physics and Biomedical Engineering, Medical University of Vienna, Waehringer Guertel 18-20, 1090 Vienna, Austria; thomas.beyer@meduniwien.ac.at (T.B.); daria.ferrara@meduniwien.ac.at (D.F.); 9Christian-Doppler Lab Applied Metabolomics (CDL AM), 1090 Vienna, Austria

**Keywords:** [^68^Ga]Ga-PSMA, PSMA PET/CT, Prostate cancer, PSA, primary tumor

## Abstract

To investigate the use of kinetic parameters derived from direct Patlak reconstructions of [^68^Ga]Ga-PSMA-11 positron emission tomography/computed tomography (PET/CT) to predict the histological grade of malignancy of the primary tumor of patients with prostate cancer (PCa). Thirteen patients (mean age 66 ± 10 years) with a primary, therapy-naïve PCa (median PSA 9.3 [range: 6.3–130 µg/L]) prior radical prostatectomy, were recruited in this exploratory prospective study. A dynamic whole-body [^68^Ga]Ga-PSMA-11 PET/CT scan was performed for all patients. Measured quantification parameters included Patlak slope (Ki: absolute rate of tracer consumption) and Patlak intercept (Vb: degree of tracer perfusion in the tumor). Additionally, the mean and maximum standardized uptake values (SUVmean and SUVmax) of the tumor were determined from a static PET 60 min post tracer injection. In every patient, initial PSA (iPSA) values that were also the PSA level at the time of the examination and final histology results with Gleason score (GS) grading were correlated with the quantitative readouts. Collectively, 20 individual malignant prostate lesions were ascertained and histologically graded for GS with ISUP classification. Six lesions were classified as ISUP 5, two as ISUP 4, eight as ISUP 3, and four as ISUP 2. In both static and dynamic PET/CT imaging, the prostate lesions could be visually distinguished from the background. The average values of the SUVmean, slope, and intercept of the background were 2.4 (±0.4), 0.015 1/min (±0.006), and 52% (±12), respectively. These were significantly lower than the corresponding parameters extracted from the prostate lesions (all *p* < 0.01). No significant differences were found between these values and the various GS and ISUP (all *p* > 0.05). Spearman correlation coefficient analysis demonstrated a strong correlation between static and dynamic PET/CT parameters (all r ≥ 0.70, *p* < 0.01). Both GS and ISUP grading revealed only weak correlations with the mean and maximum SUV and tumor-to-background ratio derived from static images and dynamic Patlak slope. The iPSA demonstrated no significant correlation with GS and ISUP grading or with dynamic and static PET parameter values. In this cohort of mainly high-risk PCa, no significant correlation between [^68^Ga]Ga-PSMA-11 perfusion and consumption and the aggressiveness of the primary tumor was observed. This suggests that the association between SUV values and GS may be more distinctive when distinguishing clinically relevant from clinically non-relevant PCa.

## 1. Introduction

Prostate cancer (PCa) is one of the most frequently reported cancers in men. Its mortality is highly dependent on the biology, molecular characteristics, and aggressiveness of the tumor cells that can be ascertained using the commonly established histopathologic Gleason scoring (GS) system and the International Society of Urological Pathology (ISUP) scores for the primary tumor [1]. The GS and ISUP classification of the primary tumor, along with the prostate-specific antigen (PSA) levels at the time of diagnosis, can predict the metastases’ presence and the likelihood of tumor recurrence. These factors provide insights into prostate tumor behavior and spread [2]. Accordingly, patients are categorized into low-, intermediate-, and high-risk clinical groups for disease recurrence, which in turn facilitates better patient management and treatment decisions. 

In this context, the role of diagnostic imaging, such as ultrasound and multi-parametric magnetic resonance imaging (mpMRI), in identifying suspicious prostate lesions is well recognized for the local staging of the tumor. Performing mpMRI prior to surgical removal of PCa allows molecular information to be obtained on whether the tumor crosses the prostatic capsule and whether it infiltrates the neurovascular bundle and neighboring organs, such as the seminal vesicle, urinary bladder, and rectum [3]. In addition, the performance of positron emission tomography (PET) molecular imaging with gallium-68 (^68^Ga) or fluorine-18 (^18^F) on prostate-specific membrane antigen (PSMA), which specifically targets PSMA peptides highly expressed on the surface of the prostate tumor cells and its accompanying metastases, allows noninvasive precise evaluation of prostate lesions. The intensity of PSMA uptake and the degree of PSMA enrichment in the tumor are reported to be positively related to Gleason grading and ISUP classification of the tumor [4,5]. Therefore, studies could prove the superiority of integrated PET/computed tomography (CT) or MRI systems for primary staging of prostate tumors in terms of diagnostic accuracy of results with fewer ambiguous findings compared with mpMRI [6]. 

However, PSMA-PET/CT or PET/MRI studies are typically performed as so-called static scans, in which PSMA uptake is measured after a predefined time following the administration of the tracer. This procedure, even though practically well aligned with clinical routine, falls short of the functional information contained in the dynamics of the tracer uptake. Dynamic studies performed over a specific region might provide more insight into the molecular entity and hallmarks of tumors than conventional static PET studies [7]. In the case of ^68^Ga-PSMA-PET/CT or PET/MRI for PCa assessment, the potential utility of dynamic PET has also been reported for monitoring therapy response [8]. 

Generally, various methods including compartmental and noncompartmental kinetic modeling are applied for the quantification of dynamic PET images [9]. Based on prior reports, ^68^Ga-PSMA uptake in primary PCa is best described by an irreversible two-tissue 3k kinetic model [10]. Therefore, Patlak analysis seems viable to process ^68^Ga-PSMA data. Patlak-based quantitative analysis, resembling the compartment model by using the Patlak slope (Ki), which indicates the absolute rate of tracer consumption, and the Patlak intercept (Vb), representing the degree of tracer perfusion in the tumor, has recently been adopted in modern integrated clinical PET/CT systems [11]. Here, the Patlak model is implemented into the reconstruction software, thus allowing for the routine use of simplified kinetic modeling in clinical practice without the need for extensive post-processing. 

Nevertheless, the clinical benefit of assessment of the direct parametric reconstruction was hitherto not fully understood. Therefore, in this study, we were interested in determining whether the use of kinetic parameters derived from direct Patlak reconstructions of [^68^Ga]Ga-PSMA-11 PET/CT are superior to the standard static parameters, to standardized uptake values (SUV), and in predicting the histological grading of malignancy of the primary tumor of patients with PCa.

## 2. Results

As depicted in Table 1, participants had a mean age of 66 (±10 years) and a median initial PSA (iPSA) value, which corresponds to the PSA level at the time of the [^68^Ga]Ga-PSMA-11 PET/CT examination, 9.3 (range 6.31–130 µg/L). Only 5 out of 13 patients had unifocal tumors confined to one side of the prostate. The rest of the patients (n = 8) presented with multifocal bilateral prostate tumors. Based on histology findings and GS grading of the prostate tumor, 7 of 13 patients were classified as having high-risk PCa (GS 8–10 and ISUP 4–5) and 5 patients had intermediate, unfavorable PCa with GS 7b (4 + 3) and ISUP 3. Only one patient had intermediate favorable risk PCa with a GS of 7a (3 + 4) and an ISUP classification of 2. A total of 20 individual malignant prostate lesions were ascertained in these 13 patients and were histologically graded for GS with ISUP classification. In total, six lesions were classified as ISUP 5, two as ISUP 4, and eight as ISUP 3, whereas four lesions were classified as ISUP 2 (Table 2).

### Quantification of the PET Data

In both static and dynamic PET imaging, it was visually possible to delineate the prostate tumor from the background. The lesion-based quantifications of the static and dynamic [^68^Ga]Ga-PSMA-11 PET/CT images of each individual prostate lesion in all participants prior to their RP are presented in Table 2. The average mean values of the SUV, slope, and intercept of the background (i.e., prostate tissue that did not visually show PSMA uptake) were 2.4 (±0.4), 0.015 mg/mL/s (±0.006) and 52% (±12), respectively. These were statistically significantly lower than the mean and max SUV, slope, and intercept parameters extracted from the prostate lesions (all *p* < 0.01). The values of static and dynamic [^68^Ga]Ga-PSMA-11 PET parameters for all studied lesions are summarized in Table 3. 

The maximum and the mean values for the SUV, slope, and intercept of the lesions in relation to ISUP classification are shown in Figure 1. Although lesions with ISUP 4 and 5 tended to have higher maximum values for each SUV, slope, and intercept than lesions with ISUP 2 and 3, no statistically significant differences were found between any of these values and the various ISUP classifications (all *p* > 0.05), as seen in Figure 1.

As expected, the results of Spearman’s correlation coefficient analysis demonstrated a strong correlation between the Gleason score and ISUP (r = 0.92, *p* < 0.001). Furthermore, the majority of the PET-based parameters exhibited significant correlations with each other (r ≥ 0.70, *p* < 0.01), except values of the Patlak intercept mean with values of the Patlak slope mean (*p* > 0.01). Both GS and ISUP grading had only weak to very weak correlations with mean, maximum, and TBR values derived from static PET images and dynamic Patlak slope (*p* < 0.05). However, no correlations were demonstrated between values extracted from the Patlak intercept and GS grading and ISUP classification (Figure 2 and Appendix A).

Notably, neither a significant correlation between the values of iPSA and GS and ISUP grading nor between the iPSA values and the values of the dynamic and static PET parameters could be observed.

## 3. Discussion

Here, we investigated the benefits of direct parametric reconstruction derived from direct Patlak reconstructions of [^68^Ga]Ga-PSMA-11 PET/CT over the standard static parameters, SUV, in predicting the aggressiveness of the primary tumor in patients with PCa. Indeed, earlier studies were able to identify positive associations of iPSA and GS with the intensity of the PSMA uptake in the primary prostate tumor [12,13]. They were able to detect great associations between SUV values acquired from static [^68^Ga]Ga-PSMA PET images and high Gleason pattern and pathological upgrading in patients with PCa and found that the degree of PSMA uptake in the primary tumor represents an independent risk factor for the biochemical recurrence [14,15]. 

In this context, dynamic PET scans could be useful to provide more accurate biological and pathological information about the primary tumor and to stratify individuals considered susceptible to disease recurrence [8,16]. The best model for the kinetic analysis of [^68^Ga]Ga-PSMA-11 is a 3k irreversible two-tissue compartment model. Therefore, the parametric Patlak data with an image-derived input function that closely mimics compartment modeling can be used to capture the dynamic parameters of [^68^Ga]Ga-PSMA-11 PSMA-PET, described previously in other studies [10,17]. It should be noted that after most of the tracer is bound to the PSMA receptors (k3) on the cell, the values of k4 may be relatively small and negligible compared with the other kinetic parameters, as they measure the rate of externalization of the PSMA tracer from the cell [18]. 

In fact, numerous studies have identified a positive correlation between the degree of aggressiveness of the primary prostate lesion based on the final GS grading and the parameters of both dynamic and static PSMA-PET and emphasize the potential of PSMA-PET parametric as a replacement for the histologic approach in suspicious prostate lesions [4,12,19,20,21]. However, the results of our current study demonstrated no statistically significant differences between the maximum and mean values of the SUV, slope, and intercept of the prostate malignant lesions and the various GS grading and ISUP classifications. In this regard, the study by Sachpekidis et al. in 24 patients with primary therapy-naive PCa also showed only a weak correlation between GS and SUV parameters, though > 95% of patients had PSMA-positive PCa lesions [22]. Some other previous studies have also been unable to identify a significant relationship between SUV parameters and iPSA and GS grading [23,24]. Similarly, in a study by Woythal et al. that involved 31 patients with primary PCa, no statistically significant correlation was observed between SUV parameters and mean tumor size and between SUV parameters and GS staging of the tumor [25]. Therefore, iPSA and PSMA-PET parameters might be less related to the degree of pathological aggressiveness of the prostate tumor. This was also seen in another study of 31 patients with BCR-PCa [26] and more recently in a study by Bogdanovic et al. that showed no or only a weak correlation between dynamic and static parameters derived from PSMA (^68^Ga-PSMA-11, ^18^F-PSMA-1007, and ^18^F-rhPSMA7) PET/MRI images and PSA levels and GS grading in 100 patients with primary PCa [27]. 

The differences in conclusions between studies reporting a significant correlation of tracer uptake with GS or ISUP and studies not demonstrating this may be due to slightly divergent study designs. Most tumor lesions investigated in our study were high-risk prostate lesions with high GS and ISUP classifications, whereas in studies reporting correlations between PSMA uptake and GS and ISUP, groups of benign or very low-risk prostate lesions were also included. This may play an important role in finding no differences in dynamic and static PSMA-PET parameters between aggressive and less aggressive prostate lesions. Taking the findings from this study together with reports in the literature, it seems questionable that tracer uptake can predict histological grading. However, a differentiation between low- and high-risk lesions using PSMA-PET/CT or PET/MRI seems reasonable [4,28]. 

Another important outcome of this work is the excellent correlation between SUV (SUVmean and SUVmax values) and kinetic parameters. This finding indicates that for clinical practice, the use of a static PET protocol provides similar information in primary PC than a time-expensive dynamic acquisition. 

Nevertheless, the small sample size of PCa patients investigated in this study and the lack of activity measurements in arterial blood sampling for accurate metabolite analysis and plasma-to-blood ratio correction are the study limitations that deserve special attention. Furthermore, our focus was the primary tumor of the prostate rather than the metastatic lesions, such as lymph nodes and bone lesions. Therefore, the findings of this study are solely limited to primary PCa. Lastly, it should be pointed out that the degree of uptake of other PET tracers that are not investigated in this study, such as choline or 2-deoxy-2-[^18^F]fluoro-D-glucose (FDG), may also have significant prognostic value for patients with PCa and could contribute substantially in further characterizing the metabolic behavior and aggressiveness of the prostate tumor [29,30,31]. Lesions displaying high FDG uptake but low PSMA expression are suggestive of high-risk tumors, which are typical in PCa patients with high-grade GS and high PSA levels who develop castration-resistant disease later in the course of the disease [32].

## 4. Methods and Materials

### 4.1. Patients 

In collaboration with the University Department of Urology, male patients with histologically confirmed primary prostate cancer prior to their planned radical prostatectomy (RP) were enrolled in this prospective study. All patients agreed and signed the informed consent form prior to undergoing dynamic [^68^Ga]Ga-PSMA-11 PET/CT imaging. The study was approved by the institutional ethics committee with EK-Nr. 1907/2020. 

In total, 19 patients with newly diagnosed PCa performed a dynamic [^68^Ga]Ga-PSMA-11 PET/CT examination. Among them, 3 patients were treated primarily with hormonal therapy and chemotherapy due to distant metastases that were detected during tumor staging by [^68^Ga]Ga-PSMA-11 PET/CT scan. Two patients opted for local radiotherapy rather than surgery. One patient received short-term ADT before the surgical removal of his prostate, so GS grading was not meaningful. The remaining 13 patients had successfully undergone planned RP. Of these, 12 patients acquired robot-assisted RP with pelvic lymphadenopathy, and only one patient received laparoscopic RP with nerve-sparing extended lymphadenopathy due to the advanced stage of the disease. Final GS staging and ISUP classification were documented in all these 13 patients. 

### 4.2. Dynamic PET/CT Data

Every studied participant underwent a dynamic [^68^Ga]Ga-PSMA-11 PET/CT examination on a Siemens Biograph Vision 600 PET/CT system operating with software version VB76. Patients were positioned head-first supine with their arms at their side. After a scout scan and low-dose whole-body CT for attenuation correction, the patient was positioned with the thorax in the field of view of the PET system. Simultaneously with the start of the PET examination, [^68^Ga]Ga-PSMA-11 (188 ± 36 MBq) was injected intravenously as a bolus dose. The first 6 min of the PET protocols were performed with the bed fixed over the chest to be able to extract the peak of the input function from the aorta or left ventricle. This was followed by an approximately 60 min dynamic whole-body scan composed of 14 sweeps in continuous table mode (6 sweeps with a speed of 8 mm/s followed by 8 sweeps with 5 mm/s) [33]. 

The PET data were reconstructed as follows (a) static images: The PET data of the last two sweeps, corresponding to an approximately 10 min acquisition time starting approx. 55 min post injection, were reconstructed using a 3D OP-OSEM algorithm with TOF information and PSF corrections. All corrections (attenuation, randoms, scatter) were applied, and a matrix size of 220 × 220 × 803 was used (voxel size of 3.3 × 3.3 × 2 mm^3^). (b) For dynamic reconstructions, the aorta was automatically identified on the low-dose CT and a volume of interest (VOI) was automatically placed within the aorta to extract an image-derived input function (IDIF). The correct placement of the VOI was visually verified by the radio technician and corrected if necessary. The VOI was then automatically transferred to the dynamic PET images to extract a full blood IDIF [34]. Applying this input function, parametric images were reconstructed with the nested direct Patlak reconstruction method using the raw PET data from the last 6 sweeps [35,36]. This resulted in a set of two image data sets: one containing the Patlak slope corresponding to net influx rate of PSMA (in mg/mL/min) and the other containing the Patlak intercept corresponding to the volume of distribution (Vb) (in %) [37,38]. 

### 4.3. PET Image Analysis 

The image data were transferred to a standard clinical viewing software 2.16.0.2 (Hermes Gold LX Hybrid Viewer—Hermes Medical Solutions, Stockholm, Sweden, version number 2.16.0.2). The images were first visually evaluated by an experienced nuclear medicine physician and an experienced medical physicist, both with 10 years of experience in nuclear medicine, to validate their applicability. For the semi-quantification of the [^68^Ga]Ga-PSMA-11 avid lesions, the physician delineated the lesions on the standard static images to obtain mean and maximum (max) SUV. Then, the volumes of interest were copied to the parametric images (Patlak slope and Patlak intercept images), and pixel mean and max values were extracted for all three image data sets. Additionally, a region of interest was manually drawn on the images in a prostate area that did not have PSMA uptake to obtain mean background values of the non-pathological prostate tissue in static and parametric images. Values for tumor-to-background ratio (TBR) were then calculated from the maximum values of the SUV, slope, and intercept of the lesion divided by the mean value of the background in both static and dynamic images.

### 4.4. Statistical Analysis 

Descriptive statistics for the patient population and all extracted parameters in relation to Gleason scoring are provided. Testing for statistical differences between values from lesions with different Gleason or ISUP scores was done using a Willcoxen test with *p*-values adjusted for multiple comparisons. A *p*-value of <0.05 was used as significance threshold. Further, since the distributions of most data were not linear, all extracted parameters were tested for correlations following Spearman’s correlation coefficients methods. *p*-values of 0.01 were used as significance thresholds for the correlations. Statistical analysis was performed using the software R-version 4.3.0. 

## 5. Conclusions

In this cohort of mainly high-risk PCa, the results revealed no significant correlation between [^68^Ga]Ga-PSMA-11 perfusion and consumption and the aggressiveness of the primary prostate tumor. This might indicate that the association between SUV values and histology grading may be more distinctive in distinguishing clinically relevant from clinically non-relevant PCa. 

## Figures and Tables

**Figure 1 ijms-24-13677-f001:**
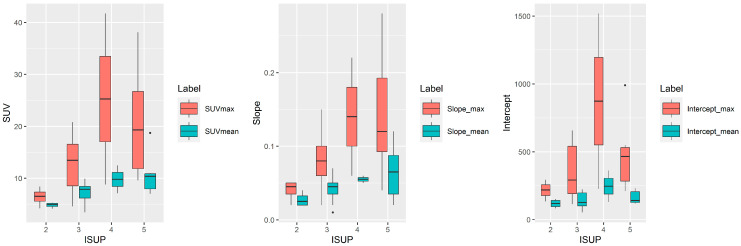
Boxplots for values of SUV, slope, and intercept of all lesions together in relation to ISUP classification. No statistically significant differences were found between values of SUV, slope, and intercept of the lesions and the various ISUP classifications (all *p* > 0.05). SUV: standardized uptake value; max.: maximum.

**Figure 2 ijms-24-13677-f002:**
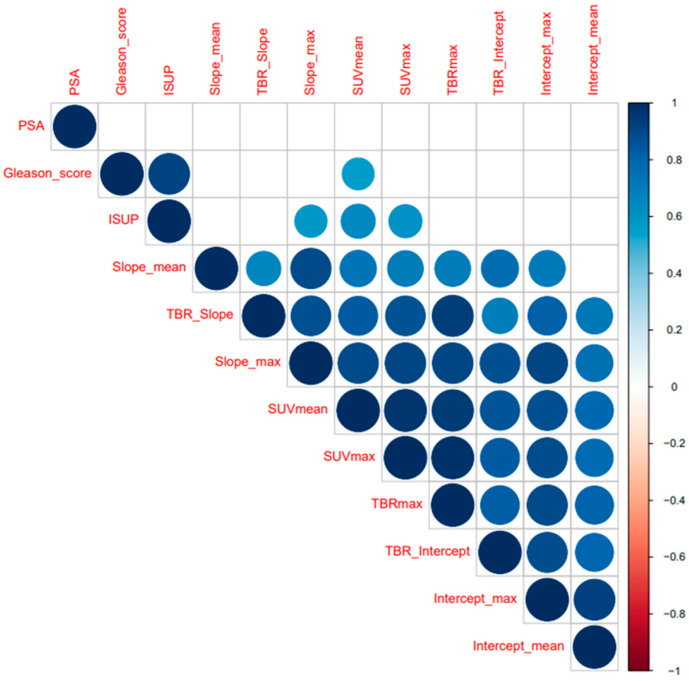
Spearman’s correlation coefficients analysis of all extracted static and dynamic imaging parameters in relation to Gleason score and ISUP grading. All static and dynamic PET parameters show significant correlations with each other (all r ≥ 0.70, *p* < 0.01), except values of Patlak intercept mean with values of Patlak slope mean (*p* > 0.01). Both GS and ISUP grading reveal only weak to very weak correlations with the mean, maximum, and TBR values derived from static PET images and from dynamic Patlak slope: *p* < 0.05. No significant associations between the values extracted from the Patlak intercept and the GS grading and ISUP classification was found. PSA: prostate-specific membrane; ISUP: International Society of Urological Pathology; max.: maximum; SUV: standardized uptake value; TBR: tumor-to-background ratio. *p*-values of 0.01 were defined as significance.

**Table 1 ijms-24-13677-t001:** Patient-based clinical and histologic characteristics of all evaluated participants with prostate cancer.

Patients	Age (Years)	Initial PSA (ng/mL)	[^68^Ga]Ga-PSMA (MBq)	Tumor Location	Gleason Score	ISUP Classification
1	74	11.9	180	bilateral	9 (4 + 5)	5
2	79	6.31	185	right side	8 (4 + 4)	4
3	52	33.9	167	right side	7 (3 + 4)	2
4	61	130	183	bilateral	9 (4 + 5)	5
5	77	7.9	300	bilateral	7 (4 + 3)	3
6	53	33	193	bilateral	7 (4 + 3)	3
7	50	12	180	left side	9 (4 + 5)	5
8	65	7.47	187	left side	7 (4 + 3)	3
9	73	6.35	134	bilateral	9 (4 + 5)	5
10	69	9.81	177	bilateral	7 (4 + 3)	3
11	61	7.2	182	bilateral	7 (4 + 3)	3
12	75	8.5	176	bilateral	9 (4 + 5)	5
13	69	9.3	202	left side	8 (4 + 4)	4

PSA: prostate-specific membrane; MBq: megabecquerel; PET: positron emission tomography; ISUP: International Society of Urological Pathology.

**Table 2 ijms-24-13677-t002:** Lesion-based parameters of static and dynamic PSMA-PET images in all participants prior radical prostatectomy.

Lesions	GS	ISUP	Static PET Images		Dynamic PET Images			
			SUVmean	SUVmax.	Patlak Slope mg/mL/min		Patlak Intercept %	
					Mean	Max.	Mean	Max.
1	9 (4 + 5)	5	10.2	14.6	0.09	0.14	120.8	450.86
2	7 (3 + 4)	2	5.1	7	0.04	0.05	137.58	243.39
3	8 (4 + 4)	4	12.45	41.7	0.06	0.22	361.65	1518.05
4	7 (3 + 4)	2	4.7	6	0.02	0.04	98.58	191.82
5	9 (4 + 5)	5	11	27.6	0.08	0.21	141.89	477.51
6	7 (4 + 3)	3	7	9.8	0.05	0.07	110.15	208.43
7	7 (4 + 3)	3	7.9	16	0.04	0.08	108.16	261.6
8	7 (3 + 4)	2	4.1	4.2	0.02	0.02	83.12	134.33
9	7 (4 + 3)	3	8.4	14.4	0.05	0.09	143.76	320.26
10	9 (4 + 5)	5	18.7	38.1	0.12	0.28	232.23	989.73
11	7 (4 + 3)	3	7.8	12.5	0.07	0.15	215.8	656.88
12	9 (4 + 5)	5	10.5	24	0.03	0.2	225.71	549.34
13	9 (4 + 5)	5	7.2	9.6	0.02	0.04	139.37	210.38
14	7 (4 + 3)	3	8.5	18.2	0.05	0.13	221.37	608.82
15	7 (3 + 4)	2	5.3	8.4	0.03	0.05	150.62	292.47
16	7 (4 + 3)	3	9.9	20.8	0.04	0.08	190.41	517.66
17	7 (4 + 3)	3	3.7	4.7	0.01	0.02	52.2	113.57
18	9 (4 + 5)	5	7	10.9	0.05	0.09	117.19	227.51
19	7 (4 + 3)	3	3.4	4.6	0.02	0.03	79.52	136.48
20	8 (4 + 4)	4	7.1	8.8	0.05	0.06	130.2	226.62

GS: Gleason score; ISUP: International Society of Urological Pathology; mg/mL/min: milligram/milliliter/minute; SUV: standardized uptake value; max.: maximum.

**Table 3 ijms-24-13677-t003:** Values of static and dynamic PSMA-PET parameters for all studied prostate lesions.

Parameters	Mean	SD	Median	Min.	Max.
SUVmax	15.1	10.7	11.7	4.2	41.7
SUVmean	8.0	3.6	7.5	3.4	18.7
Patlak slope max (mg/mL/min)	0.10	0.07	0.08	0.02	0.28
Patlak slope mean (mg/mL/min)	0.05	0.03	0.05	0.01	0.12
Patlak intercept max (%)	416.8	340.7	277.3	113.6	1518.1
Patlak intercept mean (%)	153.0	71.0	138.5	52.2	361.6
TBR SUV	6.8	6.2	4.8	1.3	24.5
TBR slope	7.6	6.8	5.0	1.0	28.0
TBR intercept	8.7	8.2	5.7	2.1	36.4

SD: standard deviation; min.: minimum; max.: maximum; SUV: standardized uptake value; TBR: tumor to background ratio.

## Data Availability

All data from this study are available to other researchers upon written request to the corresponding author.

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
