# Peer review of "Direct Patlak Reconstruction of [68Ga]Ga-PSMA PET for the Evaluation of Primary Prostate Cancer Prior Total Prostatectomy: Results of a Pilot Study"

_ijms, 2023, doi:10.3390/ijms241813677_

Round 1

Reviewer 1 Report

This is a very well presented study. 

The authors have asked an important and interesting question regarding whether dynamic uptake data of [68Ga]Ga-PSMA-11 PET could predict clinical grading of primary prostate tumours.  

The findings suggest that it cannot, at least in the small cohort of 13 participants.

The design of the study is of an exceptionally high standard and there are some important observations that may stimulate interest amongst clinician researchers, namely that the uptake signal can be seen over background earlier than possibly expected, which is useful to know in the clinical setting.

The sample size is very small and the patients are presumably from one ethnicity and cultural background.  It may not be the end of the story, but one would nevertheless expect to find stronger correlations and statistically significant data even in this limited sample if 68Ga]Ga-PSMA-11 PET was to be a powerful predictor of primary tumour grade; due to the large compensation for corrections of the parametric data in the power of the statistical tests.  The validity of the statistical approach is astutely borne out in the high correlation data between [68Ga]Ga-PSMA-11 PET uptake parameters.

Perhaps it might be useful to use 68Ga]Ga-PSMA-11 PET in combination with glucose uptake with [18F]-flurodeoxyglucose (FDG) PET PMID: 23832090. (Should it be the case that higher grade tumours will have higher metabolic and hence glucose uptake.)

There is not much more the authors can do to improve this study.  

On one hand, the whole study is interesting; on the other hand, I’m hesitant to state that it would be of significant benefit to the field as it stands alone.  

The authors have mentioned several other studies with differing results - it may be worth performing a a small meta-analysis of the relevant studies including this one to exemplify where the field currently stands on the principle topic - if it is possible to do so.  (Or otherwise there may be an alternative way to provide the reader with deeper insight into what is happening in this aspect of the field of clinical research in prostate cancer.)

Author Response

Please see the attached pdf-file. 

Reviewer 2 Report

Review for the International Journal of Molecular Sciences / IJMS (Comments to the Author)

Manuscript ID: ijms-2489411

For the paper entitled: „Direct Patlak reconstruction of [68Ga]Ga-PSMA PET for the evaluation of primary prostate cancer prior total prostatectomy: Results of a pilot study” are major and minor revisions that should be considered.

_____________________________________________________________

Only 13 patients were examined: this represents an exceptionally small number to draw conclusions from.

13 patients with 20 prostate lesions. However, in the results of the abstract, the authors list the following: 6 lesions with ISUP 5, 2 with ISUP 4, 8 with ISUP 3, 3 with ISUP 2; that makes 19 in total, but not 20.

_____________________________________________________________

Another crucial limitation is the fact that regional and distant metastases are not taken into account.

_____________________________________________________________

There is a comparable study with a similar title that the authors possibly could cite:

Direct Patlak Reconstruction From Dynamic PET Data Using the Kernel Method With MRI Information Based on Structural Similarity. IEEE Trans Med Imaging. 2018 Apr;37(4):955-965. doi: 10.1109/TMI.2017.2776324. Erratum in: IEEE Trans Med Imaging. 2018 Aug;37(8):1955. PMID: 29610074; PMCID: PMC5933939“.

_____________________________________________________________

In the abstract in purpose, the authors only write PET instead of PET/CT. However, the morphological imaging by the supplementary CT should not go unmentioned.

_____________________________________________________________

Abbreviations in the abstract must first be spelled out/defined; e.g. PET/CT.

_____________________________________________________________

If the authors state that the initial PSA values ​​were correlated, it should also be stated whether the initial PSA values ​​also agree with the PSA values ​​at the time of the examinations. There is often a longer period of time (several weeks) between the initial diagnosis by the urologist (initial PSA) and the PET/CT examination, etc.

_____________________________________________________________

The SUV (e.g. in the abstract in the results, in the discussion and the conclusion) must be specified as SUVmax, SUVmean, SUVpeak and also written out "standardized uptake value...".

_____________________________________________________________

The authors state the following in the conclusion "In this cohort of mainly high-risk PCa". It is true that 7 of the 13 patients had a high-risk PCa plus 5 of the 13 patients had an intermediate-risk PCa plus 1 patient had a PCa with a GS 7a.

In my view, the urological guidelines provide the following definitions:

GS 6 = low-risk PCa

GS 7a = low–intermediate or intermediate-favorable risk + grade group 2

GS 7b = high–intermediate or intermediate-unfavorable risk + grade group 3

GS 8 = high-risk + grade group 4

GS > 8 = high-risk + grade group 5

Therefore, any GS 7 (7a or 7b) would belong to the "intermediate-risk" PCa group (Intermediate-risk PC = GS 7 with ISUP grade groups 2 and 3). However, the authors count GS 7a as a "low-risk" PCa (e.g. in the results / 1st section on page 8).

This should be checked. If other guidelines provide for a different classification, this should be documented accordingly in the manuscript and backed up with literature.

When considering that all GS 7 patients belong to the intermediate-risk PCa, the conclusion that it is "mainly high-risk PCa" would have to be taken with caution (with 7 high-risk PCa versus 6 intermediate-risk PCa and a total of only 13 patients)!

_____________________________________________________________

In the conclusion the authors write “no correlation between [68Ga]Ga-PSMA-11 perfusion and consumption and the aggressiveness of the primary tumor was observed”. The method is not [68Ga]Ga-PSMA-11, it is [68Ga]Ga-PSMA-11 PET/CT.

Also the last paragraph on page 4 regarding 68Ga-PSMA data: [68Ga]Ga-PSMA-11 PET/CT.

In the discussion on page 11: „PSMA-PET“ and so on….

_____________________________________________________________

It would be easier for the reader to understand if nested sentence, if necessary, two sentences would be constructed:

“When combined with serum levels of prostate specific antigen (PSA) at the time of diagnosis as well the clinical local characteristics of the tumor, GS as well as ISUP classification of the primary tumor predict the presence of metastases outside the prostate and the likelihood of tumor recurrence”.

_____________________________________________________________

With regard to the information on PSMA PET imaging (top of page 4), it would be advisable to also specify hybrid imaging afterwards. In addition to the molecular and functional imaging with PSMA PET, the supplementary CT (or MRI) also provides morphological information.

And which PSMA it is.

_____________________________________________________________

In the chapter "Patients and Methods" in the subchapter "Patients" it says the following:

 „male patients with histologically confirmed primary prostate cancer prior to their planned radical prostatectomy (RP) were enrolled in this prospective study“.

However, as follows:

3 patients were treated primarily with hormonal therapy and chemotherapy due to distant metastases that have been detected during tumor staging by PSMA-PET/CT scan. Two patients opted for local radiotherapy rather than surgery. One patient received short-term ADT before the surgical removal of his prostate, so GS grading was not meaningful. The remaining 13 patients had successfully undergone planned RP. Of these, 12 patients acquired robot-assisted RP with pelvic lymphadenopathy and only one patient received laparoscopic RP with nerve-sparing extended lymphadenopathy“. Here the facts should be explained in the manuscript.

_____________________________________________________________

„The images were first visually evaluated by an experienced nuclear medicine physician“. Unfortunately, only one nuclear medicine doctor assessed the recordings, without a colleague checking them.

____________________________________________________________

My review report is attached as pdf-file.

Author Response

Please see the attached pdf-file.

Reviewer 3 Report

The Authors have investigated the use of kinetic parameters derived from direct Patlak reconstructions of Ga-PSMA PET to predict the histological grade of malignancy of the primary lesion in prostate cancer. Overall the manuscript is well written and the concept correctly expressed and scientifically demonstrated. 

However, some minor issues should be addressed:

- in the Results it is stated "no statistically significant differences were found between any of these values and the various ISUP classifications": it is true, but as it may be argued by Figure 1 there is a trend to higher SUVmax and slope_max in patients with ISUP 4 and 5 rather than in those with ISUP 3 and 3; although not significant it should be pointed out in the text

- in the results it is written that three lesions were classified as ISUP 2 but, as evident from Table 2, there are four lesions classified as ISUP 2

Best regards.

Author Response

Please see the attached pdf-file.

Round 2

Reviewer 1 Report

Thank you for revising your manuscript appropriately - perhaps you can follow it up with a review encompassing the mini-meta analysis that would provide the opportunity for a lot of interesting insights into this direction in prostate cancer.

Reviewer 2 Report

I thank the authors for their point-by-point letter.

No further comments.